# Assessing the Validity of the Past-Month, Online Canadian Diet History Questionnaire II Pre and Post Nutrition Intervention

**DOI:** 10.3390/nu12051454

**Published:** 2020-05-18

**Authors:** Justine R. Horne, Jason Gilliland, Janet Madill

**Affiliations:** 1Health and Rehabilitation Sciences, The University of Western Ontario, London, ON N6A 3K7, Canada; 2The East Elgin Family Health Team, Aylmer, ON N5H 1K9, Canada; 3Human Environments Analysis Laboratory, The University of Western Ontario, London, ON N6A 3K7, Canada; jgillila@uwo.ca; 4Department of Geography, Western University, London, ON N6A 3K7, Canada; 5School of Health Studies, Western University, London, ON N6A 3K7, Canada; 6Department of Paediatrics, Western University, London, ON N6A 3K7, Canada; 7Department of Epidemiology and Biostatistics, Western University, London, ON N6A 3K7, Canada; 8Children’s Health Research Institute, London, ON N6A 3K7, Canada; 9Lawson Health Research Institute, London, ON N6A 3K7, Canada; jmadill7@uwo.ca; 10School of Food and Nutritional Sciences, Brescia University College at The University of Western Ontario, London, ON N6A 3K7, Canada

**Keywords:** food frequency questionnaire, FFQ, CDHQII, nutrition assessment, diet history, validation, validity, dietary recall

## Abstract

Dietary intake tools are used in epidemiological and interventional studies to estimate nutritional intake. The past-month Canadian Diet History Questionnaire II (CDHQII) has not yet been validated. This study aimed to assess the validity of the CDHQII in adults by comparing dietary results from the CDHQII to the same participants’ 24-h recalls consisting of two weekdays and one weekend day. The recalls were collected using the validated multiple-pass method. Participants were asked to complete both tools at baseline, and again at 3-month follow-up. The study further aimed to determine which dietary intake tool was preferred by study participants by comparing completion rates. Data collection occurred at baseline (pre-intervention) and 3-month follow-up (post-intervention). Paired sample *t*-tests were conducted to compare means for the following nutrients (grams and %kcal): calories, protein, carbohydrates, total fat, saturated fat, unsaturated fat and sodium. Intraclass correlation coefficients of agreement and coefficients of variation were further calculated. Chi-square tests were used to determine the dietary assessment method with the greatest participant completion rate. At baseline (*n* = 104), there were no significant differences between the results of the CDHQII and three 24-h recalls (averaged), with overall moderate correlation coefficients. At 3-months (*n* = 53), there were significant differences (*p* < 0.05) between dietary intake collection methods for all nutrients assessed in this study, except for saturated fat (%kcal), unsaturated fat (%kcal), protein (%kcal) and sodium (mg). Correlation coefficients were moderate. A significantly greater proportion of participants completed the three 24-h recalls compared to the CDHQII after 3 months (completion rates of 67.2% vs. 50.8% of the sample, respectively). The CDHQII provided estimates of mean nutritional intake (calories, macronutrients and sodium) that were comparable to mean intake established from three 24-h recalls, at baseline and was validated in a sample of primarily middle-aged, college-educated, Caucasian female adults with overweight and obesity for mean baseline or cross-sectional measurement only but not for assessing individual/patient dietary intake in clinical practice (r = 0.30–0.68). This tool was not validated at 3-month follow-up. Additionally, participants preferred the three 24-h recalls to the online, past-month CDHQII.

## 1. Introduction

Dietary assessment tools are commonly used by healthcare professionals to assess the nutritional status of their patients, and by researchers in epidemiological and interventional studies to advance knowledge in nutritional science. Generally, short-term intake of food, beverages, and supplements is captured by multiple-day dietary recalls, such as three 24-h recalls [1]. To determine usual dietary intake over a longer time period, such as over one month or one year, researchers and practitioners typically employ food frequency questionnaires (FFQs) [2,3]. The decision to include multiple-day dietary recalls and/or a particular version of a FFQ in a study is based on the study objectives, hypothesis, design, and available resources, as each dietary intake method has its particular strengths and limitations [3]. Additionally, due to the nature of multiple-day dietary recalls and FFQs, there is high-risk for subjective error through both intentional and unintentional alterations in reporting [3,4]. 

Previous studies have indicated that FFQs administered online is the preferred intake method over paper-versions, and they are comparably reliable for measuring estimated dietary intakes [5]. The online and paper versions of the past-year Canadian Diet History Questionnaire II (CDHQII), adapted from the American Diet History Questionnaire II (DHQII), were analyzed for differences in mean nutrient intakes, but no significant differences were found [5]. Therefore, it was determined to be acceptable for participants to interchangeably complete their preferred version: either the online or paper version [5]. Additionally, several studies have shown that online FFQs can be used to reliably estimate dietary intakes of individual nutrients when compared to dietary recalls (≥3 days), with good reproducibility [6,7,8].

The American Diet History Questionnaire, also known as the National Cancer Institute/Block Questionnaire, has been validated in previous studies to accurately estimate dietary intakes in various populations [9,10,11,12]. The updated 152-item DHQII, based off the validated DHQ, has yet to be formally validated for the estimation of nutrient intake [13]. However, the DHQII is assumed to have similar validation findings due to very minimal updates to the nutrient database and food list used in its creation [13,14]. 

The original American Diet History Questionnaire was adapted using Canadian nutrient databases to reflect differences in American and Canadian fortification processes to create the CDHQI and CDHQII [15,16]. Furthermore, the establishment of the revised 153-item CDHQII was based on foods available to the Canadian market since 2004 and foods most commonly reported on the Canadian Community Health Survey (Cycle 2.2) [17]. Previous research revealed that the validated past-year CDHQI could accurately estimate upwards of 80% of intakes [17]. The past-year CDHQII, with an additional 19 questions, takes into account seasonality of consumption within food groups, and, includes modified questions on portion sizes to better reflect the temporary diet of the Canadian population [17].

To our knowledge, there is no existing research that compares the ***past-month*** CDHQII to any other dietary intake method, and thus, this more short-term dietary data collection tool has not yet been validated. Assessing the validity of the web-based past-month CDHQII is important in order to enhance nutrition research in Canada by potentially contributing to the number of valid tools available for collection of estimated dietary intake among participants. Based on this knowledge gap, the purpose of this study was to assess the validity of the past-month online CDHQII by comparing the nutritional results to those of three 24-h recalls (averaged) collected using the validated multiple-pass method [18], pre and post nutrition intervention. In addition, this study aims to ascertain whether participants prefer one method over the other, as determined by assessing which tool they were more likely to complete. 

## 2. Materials and Methods

The following nutritional components were compared between dietary intake tools [grams and percent of calories (%kcal) for macronutrients]: energy (calories), total fat, saturated fat, unsaturated fat, protein, carbohydrates, and sodium. The secondary objectives of this study were (a) to assess the validity prior to and following the implementation of an intervention involving training on measuring serving sizes; and (b) to determine participant preference for the past-month, online CDHQII or three 24-h recalls. The dataset from the Nutrigenomics, Overweight/Obesity and Weight Management trial was used to conduct this study. This pragmatic trial was registered with clinicaltrials.gov (NCT03015012) and was approved by the Western University Research Ethics Board (#108511). Inclusion criteria were as follows: English-speaking, enrolled in the Group Lifestyle Balance™ (GLB) weight management program, not seeing another healthcare provider for weight loss advice outside of the study, willing to undergo genetic testing, BMI ≥ 25 kg/m^2^, non-pregnant, non-lactating, and having access to a computer with email and a telephone at least one day per week. The complete NOW trial study protocol has been formerly detailed [19].

In brief, adult participants from Elgin and Middlesex Counties in Ontario were recruited from the GLB lifestyle change weight management program at the East Elgin Family Health Team (*N* = 140). Participants were primarily college-educated, middle-aged, Caucasian women with obesity (class II). Baseline data were collected during a pre-intervention two-week run-in period, and participants were asked to complete both three 24-h dietary recalls (two weekdays and one weekend day) and the past-month, online CDHQII. All participants who completed baseline dietary intake data collection participated in the trial; none were lost to follow-up during the run-in. As part of the nutrition intervention (after the run-in period), participants were asked to measure and record all of the foods and beverages they consumed for at least one week [20]. The full nutrition intervention for the Nutrigenomics, Overweight/Obesity and Weight Management trial has been previously detailed [19,21]. The first author (JH) led all nutrition interventions. 

At baseline and again at three-month follow-up data collection time points, participants were asked to complete the self-administered, semi-quantitative, 165-item, CDHQII past-month FFQ in an online format. Participants were asked about the frequency of consuming specific foods and supplements over the past month as well as questions about vegetarian diets and methods used for preparation of meat. The past-month CDHQII uses 16 categories to classify foods and beverage (e.g., fruits; rice, pasta, pizza; eggs and meat alternatives; beverages; fruits; etc.). Within each category respondents are asked more specific questions about precise products that they consume as well as their quantities and frequencies. The CDHQII uses *DietCalc* software to calculate energy and nutrient values based on responses to these questions. Participants were also asked to complete the three follow-up 24-h recalls within the same one-month period that they completed the follow-up online past-month CDHQII. This was also completed at baseline and again at 3-month follow-up. Trained research assistants collected the three 24-h recalls at baseline and three-month follow-up via telephone interviews using the validated multiple-pass method [18]. Previous research has indicated that when collecting dietary intake data using the multiple-pass method, women with obesity did not significantly over- or under-report actual dietary intake [18]. The multiple-day dietary recalls were entered into nutritional analysis software *ESHA Food Processor, Version 11.1* using the Canadian Nutrient File database. Advantages to telephone interviews for obtaining dietary intake data have been previously determined to be comparable, and, when delivered properly, have been superior to other methods of dietary data collection [22].

*IBM SPSS Statistics, Version 26.0* (© 2019 IBM Corp, Armonk NY, USA) was used to conduct paired sample *t*-tests to compare mean values for energy (calories), dietary fat, saturated fat, unsaturated fat, protein, carbohydrates, and sodium from the past-month CDHQII and three 24-h recalls. Chi-square tests were used to compare the proportion of participants completing each dietary intake tool at baseline and 3-month follow-up. Statistically significant values are determined at *p* < 0.05. Intraclass correlation coefficients of agreement as well as coefficients of variation were additionally calculated. 

## 3. Results

Table 1 and Table 2 outline the mean and standard deviations (mean ± SD) for each nutrient of interest, for the three 24-h recalls and the past-month, online CDHQII, with weak to moderate correlations at baseline and three-month follow-up. There were no significant differences at baseline between the nutritional results from the three 24-h recalls and the past-month, online CDHQII. At 3-month follow-up, significant differences were detected in all nutrients, except for %kcal from saturated fat, unsaturated fat, and protein as well as for sodium intake (mg), with weak to moderate correlations. Overall, the follow-up past-month, online CDHQII underestimated dietary intake for the nutrients of interest to the present study when compared to the three 24-h recalls collected using the validated multiple-pass method. 

A total of 140 participants met the predetermined inclusion criteria signed the informed consent form and were enrolled in the study. At baseline, there were no significant differences in the proportion of participants who completed the CDHQII vs. the three 24-h recalls. At 3-month follow-up, there was a significantly greater proportion of participants who completed the three 24-h recalls compared to the CDHQII (*p* < 0.02). Additionally, a significantly greater proportion of participants completed both dietary assessment tools at baseline compared to 3-month follow-up (*p* < 0.01). These indicators of participant preference are further detailed in Table 3 and Figure 1. 

## 4. Discussion

The results of this study have important implications for clinical practice and nutrition research. First, with significantly fewer (*p* < 0.05) participants completing the CDHQII compared to the three 24-h recalls at 3-month follow-up, it was determined that participants preferred completing the three 24-h recalls. In addition, given that 74% and 41% of participants completed *both* dietary assessment methods at baseline and 3-month follow-up, respectively, including both tools in a single study appears to lead to substantial participant burden. Therefore, when designing future studies researchers may want to consider selecting only one of these dietary intake tools, basing their decision on individualized study implications and keeping in mind participant preferences for the three 24-h recalls. This is of particular importance for reducing any risk of attrition bias in studies reporting on change in dietary intake over time. Overall, this study found that mean values established from the CDHQII, when used as a *baseline* dietary assessment tool, were not significantly different from three 24-h recalls and therefore the CDHQII is a valid tool for assessing mean nutritional values for populations as a baseline or cross-sectional measure. However, given that there were weak to moderate correlations, the CDHQII has not been validated for use on an individual level, such as for use with nutritional counseling of patients in a clinical setting. The use of the CDHQII tool for assessing follow-up dietary data (even after the provision of an intervention that included hands-on education about accurate portion sizes) should be cautioned both from an individual patient point of view and for assessing mean intakes of populations. In particular, the CDHQII significantly underestimated calories and grams of macronutrients, however it provided similar follow-up measures (compared to three 24-h recalls) for %kcal of protein, unsaturated and saturated fat, and mg of sodium. Given these findings, the past-month CDHQII may be more accurate for assessing populations’ mean dietary intake levels in observational studies with a single time-point as opposed to observational or interventional studies with multiple time points. 

In epidemiological studies, it is understood that repeated measures of estimated intakes are essential for accurately assessing dietary intake of individuals and populations, especially with a smaller sample size [1]. With repeated measures, estimated intakes tend to be higher on the first versus the consecutive administrations of web-FFQs, as described in several reproducibility studies [23,24,25,26]. In the present study, dietary intake was reported to be significantly lower overall when measured by the CDHQII compared to the three 24-h recalls, thus cautioning the use of the CDHQII to assess follow-up dietary intake. We also found that participants were more likely (and thus preferred) to complete the three 24-h recalls at 3-month follow-up compared to the CDHQII. We suggest that the ease of administration of telephone interviews as well as the pleasant social engagement involved in the collection process may have contributed to the preference for the three 24-h recalls. As well, the finding that participants in this study did not prefer the online-based CDHQII may relate to the fact that a portion of study participants were older adults (over 65) who are less likely to be computer literate and can lack insights on the ease of modern technology; this can lead to exhaustion and loss of patience [27,28]. Time to complete the online CDHQII may have also contributed to our findings. Though no official testing was conducted to determine the time required to complete the CDHQII, the time commitment is assumed to be comparable to the American DHQI, which takes approximately one hour to complete, or the *past-year* CDHQII which was found to take an average of 77 ± 32 min to complete [5,29]. However, the *past-month* CDHQII likely requires slightly less time to complete than the *past-year* CDHQII given that the past-year questionnaire has 165 items vs. 153 items for the past-month CDHQII [30]. 

Our finding that the accuracy of the past-month CDHQII was diminished at 3-month follow-up is intriguing. Previous reports indicate that pre-intervention measures of dietary intake are often under-reported due to participants not yet having been trained on serving sizes [31]. This may still have been the case for the present study, though (if present) under-reporting was similar between both dietary assessment methods. Based on previous research [31], in the present study one would expect that the CDHQII validity would remain or improve post-intervention. However, as indicated above, time to completion and reduced computer literacy may have contributed to this surprising study finding. 

Various dietary assessment tools have been validated in specific populations, for specific nutrients; validation may vary from one population to another. The implications of the present study are mainly generalizable to middle-aged, college-educated, adult Caucasian women enrolled in a weight management program. However, participants were advised to keep detailed food records as a component of the GLB intervention, and therefore nutritional intake data at 3 months may have been more accurate than other study populations (in which participants did not keep food records), thus limiting generalizability. Past work by Liu et al., for example, validated a 169-item FFQ developed for adults in Newfoundland and Labrador, which therefore can be justifiably used as a dietary intake assessment method in this particular population, but should be validated for reliable use in other populations [31]. Similarly, Shatenstein et al. validated a 73-item, semi-quantitative, self-administered FFQ for a French-language population in Montreal, Quebec [32], and the Adventist Health Study-2 cohort validated a modified FFQ specifically to estimate phytosterol intake in their specific population [33]. Future studies should focus on assessing the validity of CDHQII in different samples such as samples of male participants, those with lower and higher education levels, members of the general public not enrolled in a weight management program, and others. This would help to improve the generalizability of the CDHQII for use as a valid tool for broader populations. Future validation studies are also needed in studies where validation is the primary aim. In the Nutrigenomics, Overweight/Obesity and Weight Management study, the participants also completed other data collection and interventional activities [19], which contributed to participant burden. These are limitations of the present work.

## 5. Conclusions

Overall, the present study demonstrated that the online, past-month CDHQII provides estimates of mean nutritional intake for baseline calories, macronutrients and sodium that are comparable to those determined from validated, three 24-h recalls in a sample of primarily middle-aged, college-educated, adult Caucasian women enrolled in a weight management program. Thus, this tool was validated for baseline or cross-sectional measures for calories, macronutrients and sodium, but should be cautioned for individual estimates given the weak to moderate intraclass correlation coefficients. Therefore, the use of the CDHQII for assessing individual dietary intake (e.g., in clinical practice or for use in correlational studies) is not recommended. Additionally, the tool was not validated at 3-month follow-up and therefore results from follow-up online, past-month CDHQIIs should be interpreted with caution in this population, especially for calories and grams of macronutrients.

## Figures and Tables

**Figure 1 nutrients-12-01454-f001:**
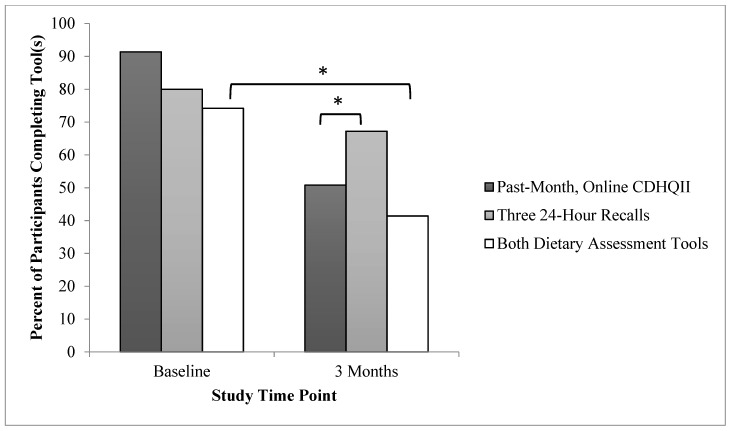
Proportion of participants completing dietary assessment tools. * *p* < 0.05.

**Table 1 nutrients-12-01454-t001:** Comparison of three 24-h recalls and CDHQII results at baseline.

	Three 24-h Recalls (Mean ± SD)	Three 24-h Recalls: Coefficient of Variation	CDHQII (Mean ± SD)	CDHQII: Coefficient of Variation	Intraclass Correlation Coefficients	Mean Differences (95% CI)	*p*-Values
Total Energy (kcal)	1773.1 ± 556.9	0.31	1702.4 ± 769.1	0.45	0.53	70.7 (−76.6 to 218.0)	0.34
Total Fat (g)	70.3 ± 28.6	0.41	68.4 ± 37.0	0.54	0.52	1.9 (−5.4 to 9.3)	0.60
Total Fat (% kcal)	35.3 ± 7.4	0.21	35.9 ± 7.4	0.21	0.55	−0.7 (−2.3 to 1.0)	0.42
Saturated Fat (g)	23.7 ± 10.4	0.44	22.5 ± 15.0	0.67	0.43	1.2 (−1.8 to 4.2)	0.43
Saturated Fat (% kcal)	11.9 ± 3.4	0.29	11.7 ± 4.4	0.38	0.30	0.2 (−0.8 to 1.2)	0.69
Unsaturated Fat (g)	45.1 ± 19.7	0.44	41.2 ± 23.0	0.56	0.52	3.9 (−0.9 to 8.6)	0.11
Unsaturated Fat (% kcal)	22.5 ± 5.5	0.24	23.1 ± 8.7	0.38	0.35	−0.6 (−2.4 to 1.2)	0.50
Protein (g)	76.5 ± 24.7	0.32	71.0 ± 39.0	0.55	0.49	5.5 (−1.8 to 12.9)	0.14
Protein (% kcal)	17.6 ± 4.1	0.23	16.7 ± 3.6	0.22	0.68	0.9 (0.2 to 1.6)	0.09
Carbohydrates (g)	200.4 ± 71.4	0.36	187.4 ± 81.1	0.43	0.60	13.0 (−2.8 to 28.8)	0.11
Carbohydrates (% kcal)	45.5 ± 9.0	0.20	44.4 ± 8.8	0.20	0.45	1.1 (−1.0 to 3.1)	0.30
Sodium (mg)	2691.3 ± 1005.3	0.37	2445.5 ± 1267.1	0.52	0.51	245.8 (−7.4 to 499.1)	0.06

Statistically significant values were evaluated using paired sample *t*-tests at *p* < 0.05 for between-group differences in means (*n* = 104).

**Table 2 nutrients-12-01454-t002:** Comparison of three 24-h recalls and CDHQII at 3-month follow-up.

	Three 24-h Recalls (Mean ± SD)	Three 24-h Recalls: Coefficient of Variation	CDHQII (Mean ± SD)	CDHQII: Coefficient of Variation	Intraclass Correlation Coefficients	Mean Differences (95% CI)	*p*-Values
Total Energy (Kcal)	**1540.0 ± 399.9**	0.26	**1291.7 ± 481.8**	0.37	0.65	248.3 (132.2 to 364.4)	**<0.01**
Total Fat (g)	**52.6 ± 20.1**	0.38	**47.2 ± 19.5**	0.41	0.66	5.4 (−0.03 to 10.9)	**0.05**
Total Fat (% kcal)	**30.3 ± 6.5**	0.21	**33.0 ± 6.2**	0.19	0.61	−2.8 (−4.5 to −1.0)	**<0.01**
Saturated Fat (g)	**16.9 ± 8.1**	0.48	**14.6 ± 6.7**	0.46	0.62	2.3 (0.2 to 4.4)	**0.03**
Saturated Fat (% kcal)	9.7 ± 3.3	0.34	10.1 ± 2.4	0.24	0.43	−0.5 (−1.4 to 0.5)	0.35
Unsaturated Fat (g)	**34.2 ± 13.3**	0.39	**28.4 ± 12.0**	0.42	0.61	5.8 (2.2 to 9.4)	**<0.01**
Unsaturated fat (% kcal)	19.6 ± 4.4	0.22	19.9 ± 5.0	0.25	0.61	−0.3 (−1.6 to 1.0)	0.64
Protein (g)	**74.5 ± 22.8**	0.31	**59.2 ± 23.3**	0.39	0.65	15.3 (9.5 to 21.1)	**<0.01**
Protein (% kcal)	19.7 ± 4.8	0.24	18.5 ± 3.6	0.19	0.60	1.2 (−0.1 to 2.4)	0.06
Carbohydrates (g)	**186.1 ± 56.1**	0.30	**147.6 ± 68.3**	0.46	0.61	38.5 (21.6 to 55.3)	**<0.01**
Carbohydrates (% kcal)	**49.0 ± 8.4**	0.17	**45.5 ± 7.7**	0.17	0.55	3.1 (0.6 to 5.5)	**0.01**
Sodium (mg)	2150.2 ± 765.4	0.36	1973.4 ± 752.0	0.38	0.50	176.7 (−63.8 to 417.3)	0.15

Statistically significant values are indicated in bold and were evaluated using paired sample *t*-tests at *p* < 0.05 for between-group differences in means (*n* = 53).

**Table 3 nutrients-12-01454-t003:** Proportion of participants completing dietary assessment tools.

	Completed Baseline CDHQII	Completed Baseline Three x 24-h Recalls	Completed 3-Month CDHQII	Completed 3-Month Three x 24-h Recalls
Sample Size (*n*; % retention)	128; 91.4	112; 80.0	**65; 50.8 ^a^**	**86; 67.2 ^a^**
	**104; 74.2 ^b^**		**53; 41.4 ^b^**	

Percent retention calculated based on *N* = 140 (participants enrolled in the study) at baseline, and *n* = 128 (participants with baseline data and therefore deemed to be remaining in the current study) at 3-month follow-up; Statistically significant values are indicated in bold and were evaluated at *p* < 0.05 (^a^
*p* < 0.02, ^b^
*p* <0.01).

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
