# Peer review of "Assessing the Validity of the Past-Month, Online Canadian Diet History Questionnaire II Pre and Post Nutrition Intervention"

_nutrients, 2020, doi:10.3390/nu12051454_

Round 1

Reviewer 1 Report

The authors of the paper validate the CDHHQ questionnaire with the previously validated 3-day, 24-hour recalls (2 working days and 1 weekend day), in a group of obese or overweight adult women. The paper is well designed and conducted. Initially, tools present a good correlation, correlation coefficients were weak to moderate, rather weak in my opinion, but it fails to maintain it after the intervention, as expected. It also assesses whether it persists after 3 months of follow-up of an intervention and patient preferences between the two tools. This article is of clinical importance, since the most adequate and precise tool to assess energy and macronutrient consumption in the population remains to be defined.

Although the new version of the article and the provision of data have improved substantially, I still believe that some aspects can be improved.

  1. Abstract.This section requires providing more information. The numbers of subjects included in each study are not specified and the numerical results are scarce, since, for example, correlation coefficients were weak to moderate.

  1. The correlation coefficient is a good measure to estimate agreement. To estimate dispersion and reproducibility, it is essential to supply the data data on inter-individual and intra-individual variability, expressed as a coefficient of variation. The most important aspect to use one or other tool is to know its reproducibility.
  2. Tables 1 and 2 only show the means and SD of the values ​​obtained with both tools, in the women to whom each tool is applied. Paired sample t-tests were conducted to compare means. It would be useful, and more informative, to know the means of the differences and their 95% CI. It would be a more precise and adequate way to report the results.

Author Response

The authors of the paper validate the CDHHQ questionnaire with the previously validated 3-day, 24-hour recalls (2 working days and 1 weekend day), in a group of obese or overweight adult women. The paper is well designed and conducted. Initially, tools present a good correlation, correlation coefficients were weak to moderate, rather weak in my opinion, but it fails to maintain it after the intervention, as expected. It also assesses whether it persists after 3 months of follow-up of an intervention and patient preferences between the two tools. This article is of clinical importance, since the most adequate and precise tool to assess energy and macronutrient consumption in the population remains to be defined.

Although the new version of the article and the provision of data have improved substantially, I still believe that some aspects can be improved.

  1. Abstract.This section requires providing more information. The numbers of subjects included in each study are not specified and the numerical results are scarce, since, for example, correlation coefficients were weak to moderate.

 Response: Please note that given another reviewer’s recommendation to replace the Pearson correlation coefficients with intraclass correlation coefficients of agreement (further reading of interest here: Koo TK and Li MY. A guideline of selecting and reporting intraclass correlation coefficients for reliability research. JCM. 2016;15), the more accurate/appropriate intraclass correlation coefficients demonstrate moderate correlations. The manuscript and abstract have been updated accordingly to reflect this change. More numerical results have been added to the abstract, including the number of participants. We could not include all numerical results given the journal’s strict limitations on the abstract’s word count.

  1. The correlation coefficient is a good measure to estimate agreement. To estimate dispersion and reproducibility, it is essential to supply the data data on inter-individual and intra-individual variability, expressed as a coefficient of variation. The most important aspect to use one or other tool is to know its reproducibility.

Response: Coefficients of variation have been added to tables 1 and 2. We have also added intraclass correlation coefficients of agreement and have updated the Methods section to reflect this revision.

  1. Tables 1 and 2 only show the means and SD of the values ​​obtained with both tools, in the women to whom each tool is applied. Paired sample t-tests were conducted to compare means. It would be useful, and more informative, to know the means of the differences and their 95% CI. It would be a more precise and adequate way to report the results.

Response: The means of the differences and their 95% CI have been added to tables 1 and 2.

We would like to thank you kindly for taking the time to review our manuscript. 

Reviewer 2 Report

Line 38: the word validated is a complex term which tends to confer an imprimatur on any subsequent use of the instrument. This is a concern for dietary intake assessment research because it leads to unwarranted invalid use of the instrument. See for example: Best Practices for Conducting and Interpreting Studies to Validate Self-Report Dietary Assessment Methods. Kirkpatrick SI, Baranowski T, Subar AF, Tooze JA, Frongillo EA.J Acad Nutr Diet. 2019 Nov;119(11):1801-1816. doi: 10.1016/j.jand.2019.06.010. Epub 2019 Sep 11. These authors advocate a more nuanced approach to reporting what  methods were used to constitute a statement of validity including the criterion used to assess validity, the validity coefficient and the sample in which the data were collected. Thus the sentence with line 38 might be reworded as: The CDHQII was tested for validity against three 24hdr at baseline and three months later showing no statistically differences in mean estimates of intake at baseline (but a tendency toward under-reporting), but statistically significant underreporting three months later with low correlations between measures at both baseline (r=0.18 to 0.53)) and three months later (r= 0.18 to 0.53), among mostly middle aged, college educated, adult white and obese Canadian women enrolled in a weight management program. The latter may limit generalizability of the findings, and suggests the CDHQII can be used to estimate group intakes at baseline, but not later, and not individual estimates or for use in correlational studies, among comparable samples.

Other uses of the word valid, validity or validated should be carefully assessed and reported.

Most studies show that the obese tend to underestimate dietary intake, even at what is here called baseline. What happened in this study to change that?

Line 115: don’t all intervention participants receive nutrition education and the control not, or different kinds of education for the two groups? So what could it possibly mean that the person providing the nutrition education was blinded?

A first table is needed which reports the demographic characteristics of the sample.

The correlations in Tables 1 and 2 should be ICCs of agreement and the method of assessing correlation should be reported in the last paragraph under Methods.

The reporting in lines 142-6 is redundant with that on 154-6. The redundancy should be eliminated.

Author Response

Recommendation: Line 38: the word validated is a complex term which tends to confer an imprimatur on any subsequent use of the instrument. This is a concern for dietary intake assessment research because it leads to unwarranted invalid use of the instrument. See for example: Best Practices for Conducting and Interpreting Studies to Validate Self-Report Dietary Assessment Methods. Kirkpatrick SI, Baranowski T, Subar AF, Tooze JA, Frongillo EA.J Acad Nutr Diet. 2019 Nov;119(11):1801-1816. doi: 10.1016/j.jand.2019.06.010. Epub 2019 Sep 11. These authors advocate a more nuanced approach to reporting what  methods were used to constitute a statement of validity including the criterion used to assess validity, the validity coefficient and the sample in which the data were collected. Thus the sentence with line 38 might be reworded as: The CDHQII was tested for validity against three 24hdr at baseline and three months later showing no statistically differences in mean estimates of intake at baseline (but a tendency toward under-reporting), but statistically significant underreporting three months later with low correlations between measures at both baseline (r=0.18 to 0.53)) and three months later (r= 0.18 to 0.53), among mostly middle aged, college educated, adult white and obese Canadian women enrolled in a weight management program. The latter may limit generalizability of the findings, and suggests the CDHQII can be used to estimate group intakes at baseline, but not later, and not individual estimates or for use in correlational studies, among comparable samples.

Other uses of the word valid, validity or validated should be carefully assessed and reported.

Response: Thank you for taking the time to re-review our manuscript. All changes have been tracked in the resubmitted document. Please note that given the word limitations for the abstract, we were not able to include your suggested statement, word-for-word, but we have re-read the article by Kirkpatrick and colleagues and have made several revisions, which include:

  • Specifying that the results of the study are generalizable only to the specific nutrients of interest in the abstract results
  • providing more details about the characteristics of the sample in the abstract
  • adding detail to the Discussion section on the generalizability of the findings (which includes your recommended additions, listed in the paragraph above)
  • adding r values to the abstract
  • calculating the ICC and adding this data to the results (in lieu of the Pearson Correlation Coefficient, which is less appropriate)

Recommendation: Most studies show that the obese tend to underestimate dietary intake, even at what is here called baseline. What happened in this study to change that?

 Response: This is why we selected the multiple-pass method as a comparison for the CDHQII. As you can see in the Conway et al. article (in which actual intake was compared to reported intake), women with obesity did not significantly over or under estimate dietary intake, when assess using the multiple-pass method. We have specified some detail about this in the Methods section. Conway, J. M.; Ingwersen, L. A. Effectiveness of the US Department of Agriculture 5-step multiple-pass method in assessing food intake in obese and nonobese women. Am. J. Clin. Nutr 2003, 77, 1176-1178.

Recommendation: Line 115: don’t all intervention participants receive nutrition education and the control not, or different kinds of education for the two groups? So what could it possibly mean that the person providing the nutrition education was blinded?

 Response: A good question, that certainly requires clarification - We have ‘blinded’ the authors’ initials (i.e. of their first and last name) for the blinded peer review process of this manuscript. If this paper is published, the author’s initials will then be inserted in place of the word “blinded.” This author was not blinded in the research sense of the term; as you indicate, this would not make sense.

Recommendation: A first table is needed which reports the demographic characteristics of the sample.

Response: The demographic characteristics of the sample have been reported in a NOW trial manuscript currently in press and therefore this data cannot be re-published in the present manuscript submitted to Nutrients. As such, we have provided a description of the sample on page 3, lines 107-108.

Recommendation: The correlations in Tables 1 and 2 should be ICCs of agreement and the method of assessing correlation should be reported in the last paragraph under Methods.

Response: Pearson correlation coefficients have been replaced with ICCs of agreement; this has been added to the Methods section.

Recommendation: The reporting in lines 142-6 is redundant with that on 154-6. The redundancy should be eliminated.

Response: These sections are not redundant as lines 142-6 (now lines 151-6) report on differences between nutritional results (Tables 1 and 2), whereas lines 154-6 (now lines 163-9) report on the completion rates for each dietary assessment tool (Table 3).

We would like to thank you kindly for taking the time to review our manuscript. 

Reviewer 3 Report

no further comments.

Author Response

We would like to thank you kindly for taking the time to review our manuscript. 

Reviewer 4 Report

The changes introduced in the article are adequate, in line with what I suggested to the authors in my previous review.

Author Response

(The authors gave the same response as above.)

Round 2

Reviewer 2 Report

The authors were responsive to my comments. Thank you!

This manuscript is a resubmission of an earlier submission. The following is a list of the peer review reports and author responses from that submission.

Round 1

Reviewer 1 Report

The authors of the paper validate the CDHHQ questionnaire with the previously validated 3-day, 24-hour recalls (2 working days and 1 weekend day), in a group of obese or overweight adult women (140 women ???). The paper is well designed and conducted. Initially, both tools present a good correlation, but it fails to maintain it after the intervention, as expected. It also assesses whether it persists after 3 months of follow-up of an intervention (which is not specifically mentioned) and patient preferences between the two tools. This article is of clinical importance, since the most adequate and precise tool to assess energy and macronutrient consumption in the population remains to be defined. Each tool provides estimate data that generally underestimates the food consumed considered "unhealthy" and overestimates the consumption of food considered "healthy"particularly in subjects who are overweight and / or obese.
In my opinion, although this study is important, it should substantially improve the exposition of the data provided in the results to be of interest.

I have some important comments that authors should consider

Tables 1 and 2 only show the means and SD of the values ​​obtained with both tools, in the women to whom each tool is applied. It would be useful, and more informative, to know the means of the differences and their 95% CI. It would be a more precise and adequate way to report the results.

Only 53 women out of the 140 included complete the study. Would it be possible to know the data of these 53 women "adherent" to the intervention? If the data were expressed with mean difference (95% CI), only these 53 women could be used, who would be the only ones who use the 2 tools at both times, or at least both tools at each time (104 at baseline).

Three months after the intervention, there is a reduction in the declaration of energy consumption and especially of total fat consumption, with both tools. Can you provide data on body weight modification? Can you explain if these results also occur in women who have also done it at baseline? Or is it simply a consequence that there are more obese women at baseline?

It would also be important to provide data on inter-individual and intra-individual variability, expressed as a coefficient of variation. The most important aspect to use one or other tool is to know its reproducibility.

Reviewer 2 Report

While the authors address an important issue, there is a serious lack of detail in methods; lack of difference does not constitute equivalence; and despite this, the authors claim validation when so many of the variables were significantly different during a time period (previous month) for both measures. I believe the authors need to declare the FFQ was not valid, when compared to three 24hdr.

Validity is hard to establish. Correlating two self-report measures would be considered concurrent validity. Remember they share much of the same self-report error. Please see SI Kirkpatrick et al. Best practices for conducting and interpreting studies to validate self-report dietary assessment methods. J Acad Nutr Diet. 2019; 119(11):1801-16.

The lack of difference does not equal equivalence. The principles and procedures from non-equivalent research designs would appear to apply high.

The sample included participants in the NOW weight loss trial and were to keep a diary. Doesn’t this mean self-selection in part on their attention to diet, and their additional attention to diet conferred by the diary and other aspects of the trial? Thus, the findings may be biased toward higher accuracy and there are limits on generalizability.

The paper would benefit from a limitations paragraph toward the end which included these concerns.

The manuscript suffers from too many acronyms. These should be limited to perhaps three total.

Line 98: The GLB acronym appears in this line, but the meaning of GLB is specified in line 104.

Since this is a dietary intake validation paper, the authors should report the types of items and response categories used, and how they computed each of the variables being compared, e.g. energy, dietary fat, etc.

The baseline data were collected during a run-in period. This suggests there were people included in the run-in period not included in the trial. This is a possible source of participation/selection bias. The authors need to report in a table the demographic and baseline dietary intake characteristics of both those in the sample participating versus those not.

The analyses would benefit from Bland-Altman plots.

What are the sample sizes in Tables 1 and 2?

There were many statistically significant differences between the two methods as reported in Table 2. How could this be considered validated, especially when Table 2 covers the same time period (previous month) covered in both measures?

Table 3 does not clearly present the data. What is baseline three or 3-month three? What the second row of data presents is opaque to me.

There was a huge loss in sample from baseline to second CUHQH (from 140 or 128 to 65), about 40% loss. What are the biases I from this loss? Should the authors recalculate the baseline differences in variables using only cases included in the second analyses?

While testing mean differences is important, testing for correlations between methods would also provide insight into how well these methods compare.

Blaming lack of equivalence on participant burden does not ring true. The sample at time 2 participated despite perceived burden. The case of use/burden findings should be considered separately from considerations of validity.

To state that the methods are valid or equivalent for baseline assessment is misleading. The second FFQ covers the same time interval as the three 24hdr, and here there were many significant differences. The authors need to report the FFQ was not found to be valid when measured against three 24hdr.

Reviewer 3 Report

Validity of the past-month, online Canadian Diet 3 History Questionnaire II pre and post nutrition 4 intervention

Horne et al aimed to assess the validity of the past-month online CDHQII by comparing the nutritional results to those of three 24-hour recalls (averaged) collected using the validated multiple pass method], pre and post nutrition intervention. In addition, their study aims to ascertain whether participants prefer one method over the other, as determined by assessing which tool they were more likely to complete.

The paper is interesting and well-written: It disentangles an important question in nutritional researche as it may be different in observational or interventional studies, as well as the literacy of participants about modern technologies.

I have only one comment for authors. In page 5, L160-162, authors state that: “First, the results indicated that after completing both tools at baseline, the study participants preferred completing the three 24-hour recalls rather than the past-month, online CDHQII”. Table 3 and Figure 1 state that 91.4% of participants completed Baseline CDHQII and 80.0% completed Baseline Three x 24 Hour Recalls. Is this correct?

Validity of the past-month, online Canadian Diet 3 History Questionnaire II pre and post nutrition 4 intervention

Horne et al aimed to assess the validity of the past-month online CDHQII by comparing the nutritional results to those of three 24-hour recalls (averaged) collected using the validated multiple pass method], pre and post nutrition intervention. In addition, their study aims to ascertain whether participants prefer one method over the other, as determined by assessing which tool they were more likely to complete.

The paper is interesting and well-written: It disentangles an important question in nutritional researche as it may be different in observational or interventional studies, as well as the literacy of participants about modern technologies.

I have only one comment for authors. In page 5, L160-162, authors state that: “First, the results indicated that after completing both tools at baseline, the study participants preferred completing the three 24-hour recalls rather than the past-month, online CDHQII”. Table 3 and Figure 1 state that 91.4% of participants completed Baseline CDHQII and 80.0% completed Baseline Three x 24 Hour Recalls. Is this correct?

Reviewer 4 Report

Este es un manuscrito que describe un estudio sobre la validez del último mes, el cuestionario en línea de historia de la dieta canadiense II antes y después de la intervención nutricional. El manuscrito estudia a mujeres caucásicas adultas de mediana edad, con educación universitaria. Es un estudio interesante y los resultados obtenidos podrían ser una contribución real al conocimiento en las herramientas de evaluación dietética. Sin embargo, existen importantes preocupaciones metodológicas que los autores deben aclarar. El análisis es adecuado pero probablemente insuficiente y se debe proporcionar más información para mitigar las limitaciones del estudio.

Cambios principales

The stated objectives are very ambitious and are not answered with the methodology used.

Materials and Methods

The first paragraph of the methodology should be removed as it is repetitive with the end of the introduction.

The author should use Energy-adjusted intakes using the residual method, where each nutrient is regressed on total calories and the population mean was then added back to the calculated residuals, according to Willet´s method

(Willett W.C., Howe G.R., Kushi L.H. Adjustment for total energy intake in epidemiologic studies. Am. J. Clin. Nutr. 1997;65:1220S–1231S)

Furthermore, to determine the validity of the FFQ it should also could computed Pearson correlation coefficients using as reference method the average of three 24hDRs.

 Results

Por lo tanto, los resultados no responden al objetivo de evaluar la validez del CDHQII en adultos comparando los resultados dietéticos del CDHQII con los mismos retiros de 24 horas de los mismos participantes.